# Brief Relaxation Practice Induces Significantly More Prefrontal Cortex Activation during Arithmetic Tasks Comparing to Viewing Greenery Images as Revealed by Functional Near-Infrared Spectroscopy (fNIRS)

**DOI:** 10.3390/ijerph17228366

**Published:** 2020-11-12

**Authors:** Zhisong Zhang, Agnieszka Olszewska-Guizzo, Syeda Fabeha Husain, Jessica Bose, Jongkwan Choi, Wanqiu Tan, Jiayun Wang, Bach Xuan Tran, Bokun Wang, Yajie Jin, Wei Xuan, Pinjia Yan, Maomao Li, Cyrus S. H. Ho, Roger Ho

**Affiliations:** 1Faculty of Education, Huaibei Normal University, Huaibei 117599, China; rsczzs@chnu.edu.cn (Z.Z.); JinYaJie@chnu.edu.cn (Y.J.); xuanwei@chnu.edu.cn (W.X.); yanpinjia@chnu.edu.cn (P.Y.); Limaomao@chnu.edu.cn (M.L.); pcmrhcm@nus.edu.sg (R.H.); 2Institute for Health Innovation and Technology (iHealthtech), National University of Singapore, Singapore 117599, Singapore; aga@nus.edu.sg (A.O.-G.); e0157451@u.nus.edu (S.F.H.); 3Department of Psychological Medicine, Yong Loo Lin School of Medicine, National University of Singapore, Singapore 119228, Singapore; e0035467@u.nus.edu; 4OBELAB, Inc., Seoul 06212, Korea; jkchoi@obelab.com; 5National University of Singapore (Chongqing) Research Institute, Chongqing 401123, China; wanqiu.tan@nusricq.cn; 6Institute for Preventive Medicine and Public Health, Hanoi Medical University, Hanoi 100000, Vietnam; bach.ipmph@gmail.com; 7Johns Hopkins Bloomberg School of Public Health, Baltimore, Maryland, MD 21205, USA; 8Science and Technology Promotion Bureau, Chongqing Liangjiang New Area Administration Committee, Chongqing 201205, China; wangbokun@lj-fund.com; 9Department of Psychological Medicine, National University Hospital, Singapore 119074, Singapore; su_hui_ho@nuhs.edu.sg

**Keywords:** arithmetic tasks, frontal cortex, functional near-infrared spectroscopy, greenery images, brief relaxation practice

## Abstract

Background: There is little understanding on how brief relaxation practice and viewing greenery images would affect brain responses during cognitive tasks. In the present study, we examined the variation in brain activation of the prefrontal cortex during arithmetic tasks before and after viewing greenery images, brief relaxation practice, and control task using functional near-infrared spectroscopy (fNIRS). Method: This randomized controlled study examined the activation patterns of the prefrontal cortex (PFC) in three groups of research participants who were exposed to viewing greenery images (*n* = 10), brief relaxation practice (*n* = 10), and control task (*n* = 11). The activation pattern of the PFC was measured pre- and post-intervention using a portable fNIRS device and reported as mean total oxygenated hemoglobin (HbO μm). Primary outcome of the study is the difference in HbO μm between post- and pre-intervention readings during a cognitive task that required the research participants to perform arithmetic calculation. Results: In terms of intervention-related differences, there was significant difference in average HbO μm when performing arithmetic tasks before and after brief relaxation practice (*p* < 0.05). There were significant increases in average HbO μm in the right frontopolar cortex (*p* = 0.029), the left frontopolar cortex (*p* = 0.01), and the left orbitofrontal cortex (*p* = 0.033) during arithmetic tasks after brief relaxation practice. In contrast, there were no significant differences in average HbO μm when performing arithmetic tasks before and after viewing greenery images (*p* > 0.05) and the control task (*p* > 0.05). Conclusion: Our preliminary findings show that brief relaxation practice but not viewing greenery images led to significant frontal lobe activation during arithmetic tasks. The present study demonstrated, for the first time, that there was an increase in activation in neuroanatomical areas including the combined effort of allocation of attentional resources, exploration, and memory performance after the brief relaxation practice. Our findings suggest the possibility that the right frontopolar cortex, the left frontopolar cortex, and the left orbitofrontal cortex may be specifically associated with the benefits of brief relaxation on the brain.

## 1. Introduction

Brief relaxation practice and viewing greenery images are recognized as non-pharmacological intervention to improve mental health. Viewing greenery images involves viewing still images of natural or vegetated elements, e.g., green spaces [1]. Viewing greenery image is not only related to the processing of perception but is also associated with emotional processing and experience [2]. Brief relaxation is provided by an audio-guide to regulate breathing patterns by slow inhalation and exhalation, leading to relaxation of the whole body in a sitting position. Brief relaxation can promote positive emotions as well as reduce anxiety and stress. Engaging in brief relaxation is relatively complex, as it involves reversing the stimulation to the autonomic nervous system and restores the body and the mind to a more balanced state. Taken together, there are clear commonalities and distinctions between brief relaxation and viewing greenery images that require further study.

Brief relaxation practice and viewing greenery images require different basic perceptuo-motor and cognitive processes. Hence, the underlying neural substrates in brief relaxation practice and viewing greenery images may be different. The neural substrates for brief relaxation practice have not been studied using traditional neuroimaging techniques such functional magnetic resonance imaging (fMRI). fMRI has been a gold standard in neuroimaging research because it provides the most accurate measure of functional activation in the whole brain. Participants have to remain absolutely still during fMRI procedures, and the slightest body movement can contribute to data errors/exclusion [2]. As a result, fMRI cannot be used during naturalistic social interactions and free limb movements due to its inability to handle motion artifacts [3].

There are only a few studies that reported functional imaging findings when a person views greenery images. In terms of regional activation, one study using fMRI found that multiple cortical regions were activated when viewing landscape gardens and natural landscapes, including the inferior occipital lobe, the left superior parietal lobule, the right fusiform gyrus, the right cuneus, and the right hippocampus [4]. Another study using functional near infrared spectroscopy (fNIRS) found that the activation for the contrast of urban versus garden showed a significant increase of oxy-hemoglobin on the right area of the prefrontal cortex. This result suggested that the garden scene might provide a pleasant and less stressful experience as compared to the urban scene for subjects [5]. This lack of functional imaging research during brief relaxation practice is not because cortical activities are not measurable, but because it would be difficult for a person to perform brief relaxation practice in a supine position in the fMRI scanner.

fNIRS is a non-invasive optical neuroimaging tool that provides robust data in the presence of movement artifacts. fNIRS has been used in various studies involving walking [6], dancing [7], as well as free arm movement [8]. fNIRS can measure the light intensity changes in near-infrared lights of wavelengths between 650 nm and 1000 nm [9]. These changes are caused by concentration variations in oxygenated hemoglobin (HbO) and deoxygenated hemoglobin (HbR) from neural activity. Viewing greenery images and brief relaxation practice both require direct attention that refers to the effortful, conscious process of bringing cognitive resources to focus on selected stimuli [10]. In our daily life, there are life events where cognitive demands exceed the mental capacity to address those demands. Based on current understanding, the frontal lobe is one of the most important neuroanatomical structures and plays key role in attention during cognitive task performance [11], emotion regulation [12], impulse control [13], and other executive functioning. A recent study measured cortical activity by electroencephalography signal when participants were exposed to park environment. This study found that frontal alpha asymmetry (FAA) values, commonly associated with positive emotions, were stronger when a person was exposed to park environment as compared to the control site [14].

The impacts of nature experience and brief relaxation practice on human cognitive function and mental health is an important research topic [10]. Hence, the present study aimed to understand the neural mechanisms on how brief relaxation practice and viewing greenery images would affect brain responses during arithmetic tasks in healthy adults. We hypothesized that there would be no difference in the oxygenated hemoglobin levels in the frontal lobe when performing arithmetic tasks after brief relaxation and viewing greenery images.

## 2. Materials and Methods

### 2.1. Participants

Thirty-one healthy subjects (mean age = 22.6 ± 1.4 years, 19 males and 12 females, 26 right-handed, 4 left-handed, and 1 ambidextrous) were recruited by word of mouth and participated in this experiment. All participants were random healthy subjects with no previous history of psychiatric, neurological, or chronic medical diseases. The participants were randomized to viewing greenery images, brief relaxation practice, or control intervention.

### 2.2. Arithmetic Task

A block design was used to assess the hemodynamic response of each participant during 3 different conditions (Figure 1). They were asked to minimize body motion during the experiment. The order of the conditions were as follows: a 1 min control period, a 1 min arithmetic period, a 3 min intervention period, and a 1 min arithmetic period. These conditions alternated with five 30 s rest periods. A fixation cross was displayed during rest periods for all participants and during intervention periods for participants randomized to brief relaxation practice and control interventions. Participants answered arithmetic questions (e.g., 6 – 4, 5 + 3) during the control and the arithmetic periods. They responded to each question by pressing a number key on a keyboard. There was no time limit per question for the control period. The time limit per question for the arithmetic period was set as 90% of each participant’s average response time. The average response for each participant was derived during a practice run in which they answered as many questions as they could in 30 s.

### 2.3. Visual Stimuli for Greenery Images and Stimulation Picture Selection

Subjects assigned to the viewing greenery intervention were asked to focus on pictures presented on the computer screen (Figure 2). During the 120 s intervention period, 9 different images were shown, and each picture was displayed for 20 s. The visual stimuli were projected onto a monitor screen (HP 23vx 23-inch Backlit Monitor, Hewlett-Packard, Palo Alto, CA, USA) in front of the participants. The viewing distance to the screen was approximately 70 cm. The size of screen was 23 inches (16:9), and the resolution was 1980 × 1080 pixels. The pictures used in this work were taken by co-author (A.O.-G.) [15]. Each of the photos in a large set was previously annotated by 10 independent landscape architecture experts with the contemplative landscape score (CLS) [15] on a 1 to 6 point Likert scale, where 1 indicates the least and 6 the most contemplative green scene. Only nine pictures with CLS equal or above 4.0 points and with good resolution were included in the study.

### 2.4. Brief Relaxation Practice

Participants assigned to the brief relaxation practice intervention were asked to familiarize themselves with the brief relaxation practice prior to fNIRS measurements. Participants were guided by verbal instructions to engage in a brief relaxation practice during the 120 s long intervention period. The instructions guided participants to focus on their breathing and bodily sensations (e.g., “Now, bring attention to your breathing, you are going to take slow, and deep breaths. Inhale, and exhale, inhale, and exhale again.”, and “Notice the feeling of your clothes on your body, your back on the chair, and your feet on the ground. As you notice these sensations, allow your mind to be calm, letting go of any stressful thoughts you may have had.”).

### 2.5. Control Intervention

Participants assigned to the control group were instructed to stare at a fixation cross for 120 s during the intervention period. The control task controlled for all aspects of the experiment except the intervention (i.e., brief relaxation practice or viewing greenery image). The control task required minimization of movement without brief relaxation practice and that the participate stare at a neutral stimulus to minimize cerebral activation and deactivation. As a result, the control task could help to demonstrate the effect on cerebral function of brief relaxation practice and viewing greenery images.

### 2.6. Measurement of Brain Activity

A portable, multi-channel continuous fNIRS imaging system (NIRSIT, two wavelengths: 780 and 850 nm; developed by the OBELAB, Seoul, Korea) with 24 laser sources and 32 photo detectors was used to measure brain activity [16]. A 48 channel probe configuration was placed on the prefrontal cortex, which allowed the mapping of each channel to the various Brodmann areas. The main neuroanatomical regions analyzed were the dorsolateral prefrontal cortex, the ventrolateral prefrontal cortex, and the frontopolar cortex [11]. The NIRSIT device received medical device approval from the Korea Food and Drug Administration in 2017. In order to minimize motion artefact and ambient light noise, the detected signals were filtered by low pass filter (discrete cosine transform (DCT) 0.05 Hz) and high pass filter (DCT 0.005 Hz) [11]. The poor quality channels decided by signal to noise ratio as 30 dB were rejected before further analysis. Hemodynamic changes for each of the 48 channels during each task were calculated by using the Modified Beer Lambert Law (MBLL) [17]. The individual results were averaged by multiple trials in each task, and grand averaging was conducted to calculate representative result in each group as expressed by mean total oxygenated hemoglobin (HbO μm) [11].

To demonstrate the differences in the prefrontal cortex activity, topographical maps were used to represent the various areas of the prefrontal cortex activation during task performance. The intensity of red color represented increased oxygenated hemoglobin levels detected in that region of the prefrontal cortex and thus correlated with increased activation and cognitive effort. The converse held true with a dark blue color representing reduced oxygenated hemoglobin levels and hypo-activation of the region. Based on the previous literature [11,18], a sample size of 30 or more was considered to be adequate for this study. To investigate the hemodynamic changes in the prefrontal cortex, the representative value (ΔHbO μm) was extracted by averaging the block averaged HbO μm in each channel based on the left and the right frontal regions. The right region refers to channels 1–16 and the left region refers to channels 33–48, as shown in Figure 3. Since the specific location of each channel in Montréal Neurological Institute (MNI) coordinates were different among participants, regional grouping could robustly calculate regional representative value at left and right frontal regions [11]. The Kolmogorov–Smirnov test and the Shapiro–Wilk test were used to test normality of (ΔHbO μm) in relaxation, viewing greenery images, and healthy control groups. Unpaired t-test was used to calculate before and after intervention differences in HbO μm when performing arithmetic tasks. All statistical analyses were performed using IBM SPSS Statistics 21 (SPSS Inc., Chicago, IL, USA). The criterion for statistical significance was set at *p* < 0.05.

The experiment was conducted upon the approval of the National University of Singapore Institute Review Board (NUS-IRB Ref No: S-19-054). All participants were informed about the procedure and the operation of the fNIRS system prior to providing written consent. Participants provided their written consent prior to the initiation of study.

## 3. Results

Thirty-one participants were randomized to the relaxation group (*n* = 10), the viewing greenery images (*n* = 10) group, and the control group (*n* = 11). Data from one participant in the control group and two participants in the viewing greenery image group were excluded due to poor quality of data during the fNIRS recording. The results of a total of 28 participants were analyzed.

Table 1 compares the percentage of correct responses during arithmetic tasks between intervention groups using one-way ANOVA. There were no significant differences in percentage of correct responses among three groups.

Figure 4 compares the PFC activation (HbO μm) when performing arithmetic tasks between pre- and post-intervention readings in the control group. There was no significant difference in average HbO μm when performing the arithmetic tasks before and after the control intervention (*p* > 0.05). T2 = Task 2; T4 = Task 4.

Figure 5 compares the PFC activation (HbO μm) when performing arithmetic tasks between pre- and post-intervention readings in the viewing greenery image group. There was no significant difference in average HbO μm when performing the arithmetic tasks before and after viewing the greenery images (*p* > 0.05).

Figure 6 compares the PFC activation (HbO μm) when performing arithmetic tasks between pre- and post-intervention readings in the relaxation group. There was significant difference in average HbO μm when performing the arithmetic tasks before and after brief relaxation practice (*p* < 0.05). There was significant increase in average HbO μm in the right frontopolar cortex (*p* = 0.029), the left frontopolar cortex (*p* = 0.01), and the left orbitofrontal cortex (*p* = 0.033) during arithmetic tasks after brief relaxation practice.

## 4. Discussion

This study aimed to identify the functional brain changes underlying viewing greenery images and brief relaxation practice by comparing the brain activation during arithmetic tasks before and after the intervention. In contrast to our hypothesis, we found significant activation in the right frontopolar cortex, the left frontopolar cortex, and the left orbitofrontal cortex during arithmetic tasks after brief relaxation practice but not after viewing greenery images or control tasks. This study established that brief relaxation practice significantly enhanced the oxygenated haemoglobin levels in keys areas of the prefrontal cortex during arithmetic tasks. Our study may have identified fNIRS-based biomarkers to assess different psychological interventions for relaxation and mental exercises. This finding encourages students and workers to practice brief relaxation from time to time to enhance cognitive function.

After brief relaxation practice, there was significant activation in several key neuroanatomical areas. Previous research found that the right frontopolar cortex is involved in the evaluation of alternate course of action [19], resource allocation [20], direct exploration [21], visual-spatial prospective memory [22], and confidence in short-term recognition memory performance [23], the left frontopolar cortex is involved in the reallocation of attentional resources [24], and the left orbitofrontal cortex is involved in differentiating the mental world from the external world and identifying people, places, and tools [25]. The above neuroanatomical areas are common cortical regions that are activated during arithmetic tasks after brief relaxation practice. Future research is required to study the above areas as region of interests (ROI) by practicing brief relaxation in a longitudinal study.

This study has several limitations. First, the fNIRS device focused on the pre-frontal cortex and could not perform whole brain assessment and was not able to assess deeper neuroanatomical structures [26]. Second, our sample size was small but was comparable with previous fNIRS studies [11,27,28]. Third, due to limitation of time and consistency, this study only involved one cognitive task (i.e., arithmetic tasks) and did not include other cognitive tasks (e.g., verbal fluency test, trail making task), and further research is required to assess the effect brief relaxation practice on other cognitive tasks. Fourth, this study involved viewing greenery images (digital images on the screen), and our finding did not represent the direct experience in outdoor greenery environment. A previous study found that direct experience with greenery and nature might offer cognitive benefits [29]. Finally, more participants (n = 2) were excluded from the viewing greenery image group due to poor data quality, and this might have affected the results of this group.

## 5. Conclusions

The present study aimed to examine differences in activation patterns in the frontal cortex during arithmetic tasks after brief relaxation practice and viewing greenery images. In conclusion, our preliminary findings show that brief relaxation practice but not viewing greenery images led to significant frontal lobe activation during arithmetic tasks. The present study demonstrated, for the first time, that there was an increase in activation in neuroanatomical areas, including the combined effort of allocation of attentional resources, exploration, and memory performance after the brief relaxation practice. Our findings suggest the possibility that the right frontopolar cortex, the left frontopolar cortex, and the left orbitofrontal cortex may be specifically associated with the benefits of brief relaxation practice on the brain.

## Figures and Tables

**Figure 1 ijerph-17-08366-f001:**
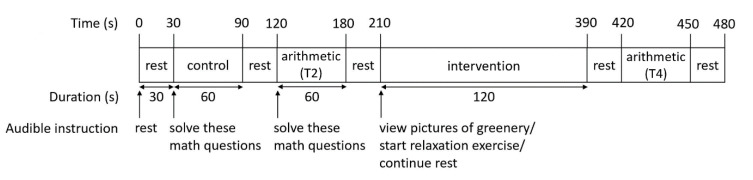
The imaging protocol and task arrangement. T2 = Task 2; T4 = Task 4.

**Figure 2 ijerph-17-08366-f002:**
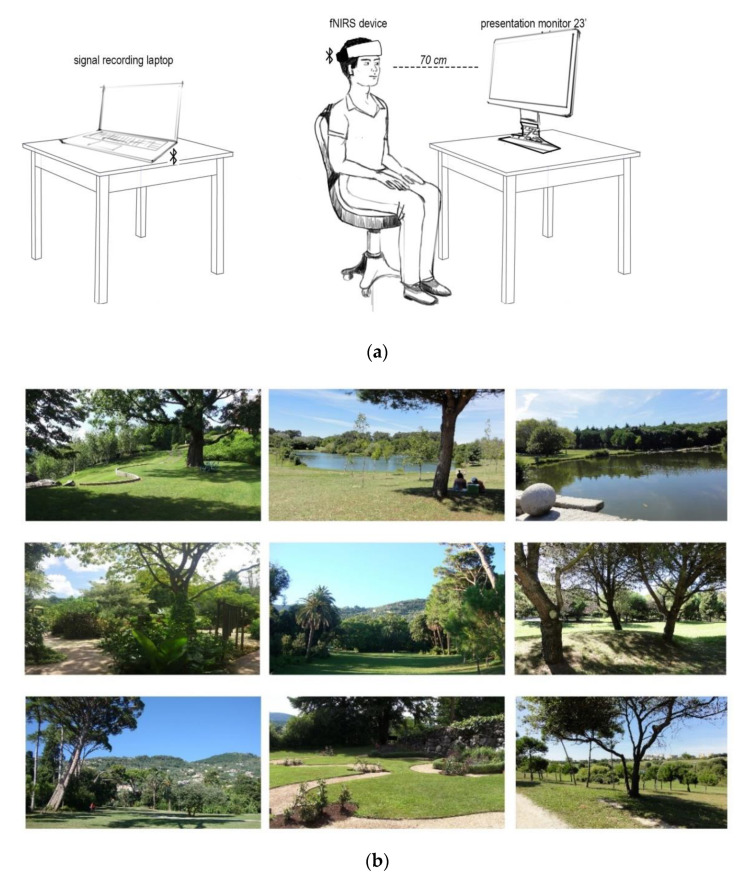
(**a**): Experiment setup with a participant, the fNIRS device and desktop monitor; (**b**): Greenery images used in the experiment, with contemplative landscape score above 4.0 points.

**Figure 3 ijerph-17-08366-f003:**
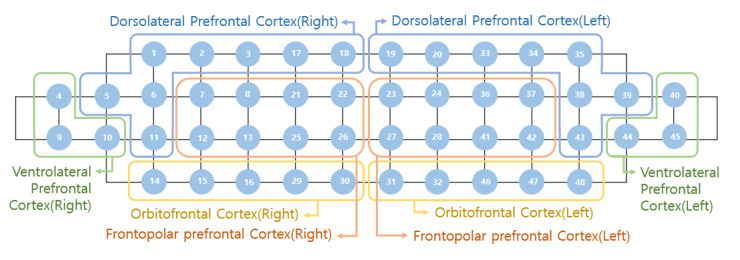
Arrangement of 48 channels and grouping of channels during data analysis.

**Figure 4 ijerph-17-08366-f004:**
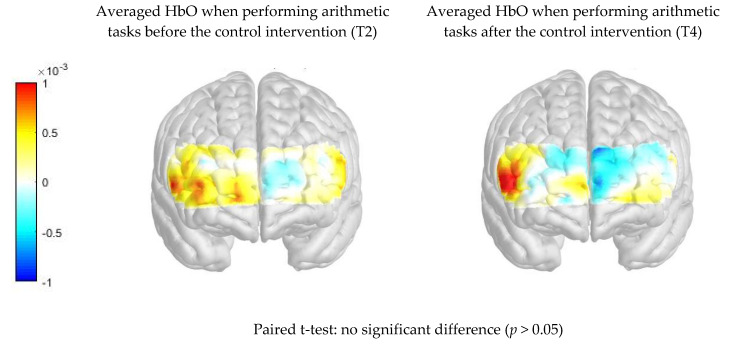
Comparison of difference in prefrontal cortex (PFC) activation (oxygenated hemoglobin (HbO) μm) when performing arithmetic tasks between pre- and post- intervention in the control group.

**Figure 5 ijerph-17-08366-f005:**
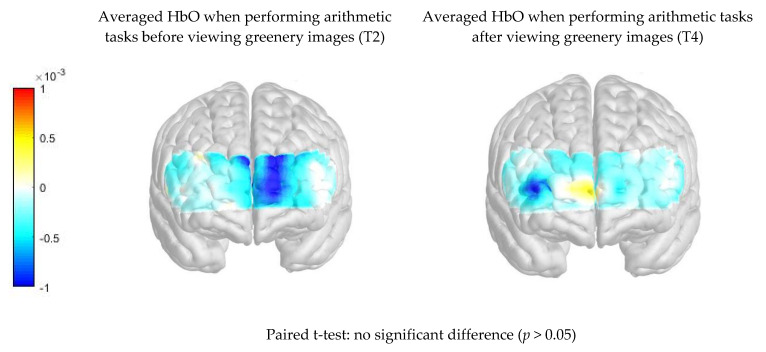
Comparison of difference in PFC activation (HbO μm) when performing arithmetic tasks between pre- and post- intervention in the viewing greenery image group.

**Figure 6 ijerph-17-08366-f006:**
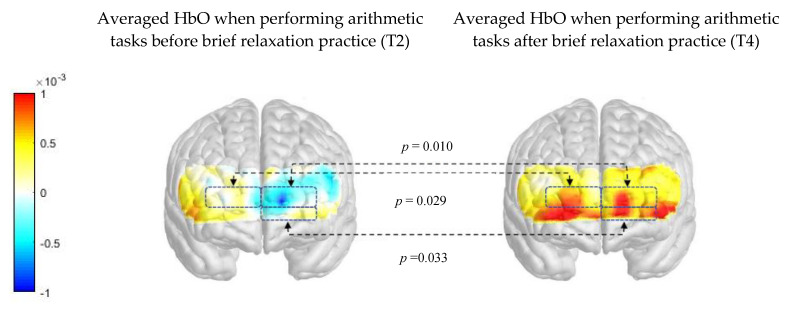
Comparison of difference in PFC activation (HbO μm) when performing arithmetic tasks between pre- and post- intervention before the brief relaxation practice.

**Table 1 ijerph-17-08366-t001:** Comparison of correct responses during arithmetic task among three groups.

	Brief Relaxation Practice (*n* = 11)	Viewing Greenery Images (*n* = 8)	Control Group (*n* = 9)	*p*-Value
Arithmetic task at T2 (% of correct responses)	81.8 ± 13.2	78.8 ± 13.2	85 ± 10.3	0.591
Arithmetic task at T4 (% of correct responses)	74 ± 18.2	77.3 ± 13.2	86.1 ± 10.9	0.202

T2 = Task 2; T4 = Task 4.

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
