# Peer review of "Brief Relaxation Practice Induces Significantly More Prefrontal Cortex Activation during Arithmetic Tasks Comparing to Viewing Greenery Images as Revealed by Functional Near-Infrared Spectroscopy (fNIRS)"

_ijerph, 2020, doi:10.3390/ijerph17228366_

Round 1

Reviewer 1 Report

The manuscript presented by Zhang and co-authors describes experiments of observation of green environments and specific breathing and relaxation exercises and their impact on the brain oxygenation. The experiments are well described, simple in a way and show interesting results. I have a few suggestions that should be addressed in the paper:

1 The first phrase of the paper can lead to misinterpretation:

  • Relaxation, in the context of imaging can mean something different to what the authors refer to, as the authors are referring to fMRI from the very beginning, it is necessary to differentiate between the relaxation as understood from MR physics.
  • Also the terms “relaxation” and “exercise” may be contradictory and can be interpreted differently to some people it may be respiration techniques to others may be a gentle stroll to others may be a leisure cycle through the woods. It would be good to clarify this from the beginning with an extra couple of phrases to define relaxation on its own, and then link it with exercise and the environment.
  • “viewing greenery images” this may be looking at images of plants, or walking in a forest. Again it would be good to clarify this, especially as the experiments will be related to viewing images

2 “fNIRS is a novel, non-invasive optical neuroimaging tool that provides robust data in the presence of movement artifacts” This phrase needs references, and I am not sure if fNIRS can be classified as novel Kwee and Nakada’s paper is from 2003.

3 Participants, were there any exclusions or pre-requisites or were these just random people?

4 A cartoon of the setting for 2.3 would be useful to visualise the participant, monitor, etc.

5 Control: This is probably out of the scope but I think that the most relevant controls would be a) images that are not of greenery, like urban settings, and b) images of green colours, but not of vegetation, like a salad. Then the comparisons would be more interesting to see if it is the colour or the environment which matter.

Author Response

Replies to Reviewer 1:

Reviewer 1: The manuscript presented by Zhang and co-authors describes experiments of observation of green environments and specific breathing and relaxation exercises and their impact on the brain oxygenation. The experiments are well described, simple in a way and show interesting results. I have a few suggestions that should be addressed in the paper:

1 The first phrase of the paper can lead to misinterpretation:

Relaxation, in the context of imaging can mean something different to what the authors refer to, as the authors are referring to fMRI from the very beginning, it is necessary to differentiate between the relaxation as understood from MR physics.

Authors: Thanks for your comments. We changed to “brief relaxation practice” as explained below. Please note that the amendments in the manuscript are coloured in red.

Reviewer 1: Also the terms “relaxation” and “exercise” may be contradictory and can be interpreted differently to some people it may be respiration techniques to others may be a gentle stroll to others may be a leisure cycle through the woods. It would be good to clarify this from the beginning with an extra couple of phrases to define relaxation on its own, and then link it with exercise and the environment.

Authors: We are sorry about the confusion. This study did not involve the environment as it was conducted in the laboratory. We have removed the statement related to the environment when discussing brief relaxation practice. We have amended the definition of brief relaxation as follows:

Brief relaxation practice is provided by an audio-guide to regulate breathing patterns by slow inhalation and exhalation, leading to relaxation of the whole body in a sitting position. Brief relaxation practice can promote positive emotions as well as reduce anxiety and stress.

Reviewer 1: “viewing greenery images” this may be looking at images of plants, or walking in a forest. Again it would be good to clarify this, especially as the experiments will be related to viewing images.

Authors: Thank for your suggestion. We have modified the definition of viewing greenery images based on an article published by IJERPH. The new definition is as follows:

Viewing greenery images involves viewing still images of natural or vegetated elements, e.g. green spaces [1]. Viewing greenery image is not only related to the processing of perception but is also associated with emotional processing and experience.

Reference:

  1. van den Berg MM, Maas J, Muller R, Braun A, Kaandorp W, van Lien R, van Poppel MN, van

Mechelen W, van den Berg AE. Autonomic Nervous System Responses to Viewing Green and Built Settings: Differentiating Between Sympathetic and Parasympathetic Activity. Int J Environ Res Public Health.12(12):15860-74.

Reviewer 1: 2 “fNIRS is a novel, non-invasive optical neuroimaging tool that provides robust data in the presence of movement artifacts” This phrase needs references, and I am not sure if fNIRS can be classified as novel Kwee and Nakada’s paper is from 2003.

Authors: Thanks for highlighting this point and we agree with your suggestion. fNIRS has been available since 2003 and it is not as novel as in the past. We have removed the word “novel” but keep non-invasive optical neuroimaging as this description still stands.

Reviewer 1: 3 Participants, were there any exclusions or pre-requisites or were these just random people?

Authors: Yes, the participants were random subjects and we have stated the exclusion criteria under methodology:

All participants were random healthy subjects with no previous history of psychiatric, neurological or chronic medical diseases.

Reviewer 1: 4 A cartoon of the setting for 2.3 would be useful to visualise the participant, monitor, etc.

Authors: Thank you for your suggestion. We have added the cartoon in Figure 2.

Reviewer 1: 5 Control: This is probably out of the scope but I think that the most relevant control task would be a) images that are not of greenery, like urban settings, and b) images of green colours, but not of vegetation, like a salad. Then the comparisons would be more interesting to see if it is the colour or the environment which matter.

Authors: Thanks for your suggestion and we provide more information about the control task. The control task controls all aspects of the experiment except the intervention (i.e. brief relaxation or viewing greenery images). For this study, the control task requires minimization of movement without brief relaxation and without exposure to other images that may trigger cerebral activation or deactivation. Then we could demonstrate the effect on cerebral function by brief relaxation and viewing greenery images. As a result, the participants who were assigned to the control group were asked to minimize movement and stare at a cross that is a neutral stimulus during the control task. We have provided more information under section 2.5.

Reviewer 2 Report

I read your manuscript with the great interest. Although it is regrettable that your hypothesis did not prove, an accumulation of the knowledge related to the research field is very important.

I have one question and one comment.

Question:

You mentioned “Unpaired t-test was used to calculate between-group differences.” However, your results shown in the manuscript were between pre- and post-intervention readings in each group and you did not show the results of between-group comparisons. Why?

Comment:

You hypothesized that the greenery images would enhance the activity of the frontal lobe. However, a contrary hypothesis may consist; the greenery images would calm down the activity of the frontal lobe without a fall in the performance of a cognitive task.

Author Response

Replies to Reviewer 2:

Reviewer 2: I read your manuscript with the great interest. Although it is regrettable that your hypothesis did not prove, an accumulation of the knowledge related to the research field is very important. I have one question and one comment.

Question: You mentioned “Unpaired t-test was used to calculate between-group differences.” However, your results shown in the manuscript were between pre- and post-intervention readings in each group and you did not show the results of between-group comparisons. Why?

Authors: Thank you for highlighting the mistake. We have made the correction as follows (coloured in red under section 2.6:

Unpaired t-test was used to calculate before and after intervention differences in HbO2μm when performing arithmetic task.

Reviewer 2:  You hypothesized that the greenery images would enhance the activity of the frontal lobe. However, a contrary hypothesis may consist; the greenery images would calm down the activity of the frontal lobe without a fall in the performance of a cognitive task.

Authors: Thank you for your comments. In order to make it clearer, we have rephrased the hypothesis to state the null hypothesis as follows:

We hypothesized that there would be no difference in the oxygenated hemoglobin levels in the frontal lobe when performing arithmetic task after brief relaxation and viewing greenery images.

For this study, there was a control group which did not expose to brief relaxation and viewing of greenery images. We found that viewing greenery images had similar findings in terms of oxygenated hemoglobin levels during the arithmetic task as the control group. Regarding cognitive performance, this study also measured the percentage of correct responses during arithmetic tasks and there was no difference among the three groups. We have added the following table in the manuscript under the results.

Table 1 compares the percentage of correct responses during arithmetic tasks between intervention groups using one-way ANOVA. There were no significant differences in the percentage of correct responses among the three groups.

Table 1.  Comparison of correct responses during arithmetic task among three groups

Brief relaxation (n=11)

Viewing greenery images (n=8)

Control group (n=9)

p-value

Arithmetic task at T2 (% of correct responses)

81.8 ± 13.2

78.8 ± 13.2

85 ± 10.3

0.591

Arithmetic task at T4 (% of correct responses)

74 ± 18.2

77.3 ± 13.2

86.1 ± 10.9

0.202